# Unmet Mental Health Care Needs among Asian Americans 10–11 Years After Exposure to the World Trade Center Attack

**DOI:** 10.3390/ijerph16071302

**Published:** 2019-04-11

**Authors:** Winnie W. Kung, Xiaoran Wang, Xinhua Liu, Emily Goldmann, Debbie Huang

**Affiliations:** 1Fordham University, Graduate School of Social Service, 113 West 60th Street, New York, NY 11375, USA; xwang150@fordham.edu; 2Columbia University, Mailman School of Public Health, 722 W 168th Street, New York, NY 10032, USA; xl26@cumc.columbia.edu (X.L.); dh2652@cumc.columbia.edu (D.H.); 3New York University, College of Global Public Health, 715/719 Broadway, 10th floor, New York, NY 10003, USA; esg236@nyu.edu

**Keywords:** unmet mental health care needs, Asian Americans, World Trade Center attack, disaster, mental health conditions, mental health service use, health insurance, social support, stressful life events

## Abstract

This study investigated the prevalence of unmet mental health care needs (UMHCN) and their associated factors among 2344 Asian Americans directly exposed to the World Trade Center (WTC) attack 10–11 years afterwards. Given the pervasive underutilization of mental health services among Asians, their subjective evaluation of unmet needs could provide more nuanced information on disparities of service. We used the WTC Health Registry data and found that 12% of Asian Americans indicated UMHCN: 69% attributing it to attitudinal barriers, 36% to cost barriers, and 29% to access barriers. Among all the factors significantly related to UMHCN in the logistic model, disruption of health insurance in the past year had the largest odds ratio (OR = 2.37, 95% confidence interval: 1.61–3.48), though similar to functional impairment due to mental disorders. Post-9/11 mental health diagnosis, probable mental disorder and ≥14 poor mental health days in the past month were also associated with greater odds of UMHCN, while greater social support was associated with lower odds. Results suggest that continued outreach efforts to provide mental health education to Asian communities to increase knowledge about mental illness and treatment options, reduce stigmatization of mental illness, and offer free mental health services are crucial to address UMHCN.

## 1. Introduction

Numerous studies have examined the mental health impact of the September 11, 2001 attack at the World Trade Center (WTC) on those who were directly exposed to the disaster, but fewer have investigated these individuals’ use of mental health services [1,2]. While Asian Americans constituted a sizeable number of those affected by the disaster [3] and there has been a well-documented history of underutilization of mental health services within this population [4], our team was the first to study this group’s mental health service use and its associated factors (the authors, in press). In this study, we further investigate the unmet mental health care needs (UMHCN) of Asian Americans affected by the WTC attack and correlates of having unmet needs.

Mental health service utilization is a typical measure of whether mental health care needs are met among persons with potential mental health issues [5]. It has been used as an objective measure to detect systemic disparities in service provision and access across sociodemographic groups [6]. However, we cannot assume that receiving services is equivalent to having mental health needs met [5,7]. Individuals’ subjective perception of UMHCN is an important measure to complement the objective information on treatment use and reveals the “actual demand for service” [7] or treatment gap. According to the National Comorbidity Survey-Replication, help seeking for mental health treatment is often delayed; for example, studies have noted a median lag of 12 years between the onset of posttraumatic stress disorder (PTSD) and the initial treatment contact [8]. Together with the often delayed symptom manifestation and worsening of symptoms with time [9], investigation of UMHCN some years after the WTC attack is important. This study examines the UMHCN of Asian Americans 10–11 years after the disaster. We aimed to better understand the extent of such unmet needs and their associated factors to inform outreach efforts in this underserved population and to find ways to provide more effective services to meet their needs. This is especially relevant given the provision of public funds to monitor and provide needed health and mental health treatment to eligible individuals affected by the WTC attack through the James Zadroga 9/11 Health and Compensation Act being extended until 2090 [10]. Meeting such unmet mental health needs may not only reduce human suffering but also ameliorate economic loss that incurs to society due to compromised productivity and increased future treatment costs when untreated conditions deteriorate [11].

UMHCN may be more common among individuals with specific clinical, sociodemographic, and behavioral characteristics. Individuals who have more mental health issues, including those with formal diagnoses or symptoms of probable mental disorder, may have more mental health care needs and are thus more likely to feel such needs not being met. However, since UMHCN captures the subjective perception of these needs, those who do not have a formal or probable diagnosis but feel that their needs have not been met should also be considered [5]. Furthermore, many individuals without apparent mental disorders also consume services [12]. The fact that the majority of ethnic and racial minorities with a diagnosable mental disorder did not receive pertinent treatment despite contacts with the health care system in the previous year [12] and the questionable validity and reliability of mental health assessment tools for minority groups [13] increase the need to examine Asian Americans without formal clinical evaluation to understand their subjective UMHCN. Poorer perception of mental well-being and greater subsequent impaired functioning in work, family or social life may also cause them to recognize that their mental health needs are not being met. Functional impairment may be even more important than mental health symptoms themselves among Asians, who tend to minimize their psychological distress [14], particularly for those who have fewer resources to sustain themselves and their families when they cannot perform their expected roles, especially work role, which is highly valued among Asians [15]. 

Limited economic resources such as low or unreliable income and the lack or disruption of health insurance coverage may also be practical barriers that could result in UMHCN [6]. Likewise, access barriers such as unavailability of culturally-relevant and linguistically-appropriate mental health services, lack of knowledge of available services, or inconvenience in service use like distance and long waiting periods may also lead to UMHCN [16]. These practical barriers may have a particularly strong impact on Asian Americans given their lower prevalence of English proficiency (76.5%) [17] and shortage of bilingual mental health services [18,19]. 

In addition to practical barriers, UMHCN can also be a result of “subjective chosen unmet needs” [6], in which individuals with mental health issues refuse to seek help due to beliefs such as the fear of stigma associated with being identified as mentally ill or needing psychiatric help [20,21]. These attitudinal or emotional barriers may be aggravated among Asians due to their cultural tendency to link mental illness to moral failure such as punishment by God or ancestors for one’s past transgressions [22] and weakness in spiritual strength, causing shame and disgrace to family [23]. As a result, the need to seek treatment could pose a threat to one’s self-esteem [24,25]. 

Social support resources may also determine UMHCN. Social networks can provide emotional support, which may buffer the psychological impact of trauma [26] and subsequently reduce the need for professional help [27]. These networks can also provide practical support through sharing of information about available mental health services and alleviating access barriers, such as providing child care [5]. Asians’ tendency to handle personal issues within close circles of family and friends may reduce their perceived need for help from outsiders, especially professionals [13,28]. Moreover, the social network’s attitude towards mental health issues and treatment could also aggravate the fear of stigma associated with using external help [25]. Further, those who have mental health issues but are unaware of such a need would understate their unmet need [6,29]. It was noted that minorities, including Asians, with limited English proficiency tend to be less likely to identify mental health care needs [30]. 

Not only could individuals who have received mental health treatment still find their needs unmet, prior studies, including those on the WTC attack, consistently find that individuals who have used services in the past are more likely to report UMHCN [2,5,7,27]. It is suggested that those who have received treatment before are more aware of the range of potential help to which they have no access, or they may better recognize the limitations of these services [5]. Those with prior experience with mental health services may also express greater dissatisfaction with the quality of treatment they receive [7]. Minority populations, in particular, may be more likely to receive inconsistent diagnoses or inappropriate medication, or be offered treatments that are not evidence-based [13,31]. Service users may also judge treatments to be inadequate when their expectations are not met. If their expectations are unrealistic, such as a quick fix, they can be corrected through good communication between service providers and consumers. However, for many minorities, including Asians, this is harder to attain due to the different socioeconomic, cultural, and linguistic backgrounds between consumers and service providers [32,33,34]. Studies indicate that minorities often find that their service providers do not listen to or understand them [35,36]. For mental health treatment to be effective, verbal communication between recipients and providers is of paramount importance so that consumers’ subjective experience can be understood and providers can effectively monitor and provide treatment. This applies not only to counseling where a therapeutic alliance is the vehicle for change [37,38] but also for pharmacological treatment, in which trust is often tied to medication compliance [39]. When discrimination is experienced by minority service recipients, including Asians [26,40], such negative encounters can leave them feeling that their mental health needs have not been met. 

The underutilization of mental health service among Asian Americans has been well documented for decades [4] and was noted to have persisted in more recent reviews [34]. However, it is possible that the widely publicized and acknowledged trauma of the WTC attack could reduce the stigma of mental illness, thereby normalizing the need for mental health service and increasing awareness of unmet needs. This study aimed to examine the prevalence and factors associated with UMHCN among this understudied and marginalized Asian American population 10–11 years after the disaster.

## 2. Methods

### 2.1. Study Participants

The World Trade Center Health Registry: The current study used data collected from the WTC Health Registry (hereafter referred to as the Registry). Funded by the US Federal Emergency Management Agency (FEMA), the Registry was developed in 2002 through the collaboration of the New York City Department of Health and Mental Hygiene and the Agency for Toxic Substances and Disease Registry to monitor the long-term health effects of the 9/11 attacks. Eligible participants included rescue and recovery workers, workers in the WTC and nearby buildings, passersby, and residents in lower Manhattan. Data were collected in 2003–2004 (wave 1), 2006–2007 (wave 2), 2011–2012 (wave 3), and 2015–2016 (wave 4). At wave 1, 71,437 participants age 18 or above were recruited. Full details of the recruitment and data collection processes have been described elsewhere [41,42].

Current Study Participants: To understand longer-term UMHCN among Asian American participants in the Registry, this study employed wave 3 data collected 10–11 years after the disaster. We included all adult participants (aged 18 or older) at wave 3 who self-identified as Asian American (*N* = 2538). We then excluded those with missing values for UMHCN and those with pre-9/11 mental health diagnosis. A total of 2344 Asians were included in the final analytic sample. Institutional Review Board approvals were obtained from the first and second authors’ affiliated institution, the Centers for Disease Control and Prevention, and the New York City Department of Health and Mental Hygiene.

### 2.2. Measures

Unmet mental health care needs (UMHCN) were the outcome of interest and were captured by asking if, in the past 12 months, the respondent “needed mental health care or counseling but didn’t receive it”. Attributions for UMHCN were divided into *attitudinal barriers*, which included endorsement of any one or more of the statements: “preferred to manage myself”, “didn’t think anything could help”, “afraid to ask for help or of what others would think” or “didn’t get around to it or didn’t bother.”; *cost barriers*, which included endorsement of “couldn’t afford to pay” or “no insurance or not covered by insurance.”; and *access barriers*, which included endorsement of any one or more of the statements: “did not know where to go or what kind of doctor to go to for care”, “problems with transportation, scheduling, childcare, or other family responsibilities”, or “was unable to find a provider who could diagnose or treat my condition.” Participants could check all that applied; thus, the three categories were not mutually exclusive. Sociodemographic variables included gender (female or male), education (college graduate or college graduate/higher, collected at wave 1), age (18–24, 25–44, 45–64, 65+, calculated at wave 3), marital status (married/living with partner, divorced/separated/widowed, or never married, collected at wave 3), household income, nativity, and employment status (collected at wave 3). WTC attack exposure quantified direct disaster exposure and was measured as the number of distinct disaster experiences endorsed (collected in waves 1 and 2), including having been in a damaged/collapsed building during the attack, witnessing three or more horrific events (e.g., saw a plane hit a tower), experiencing intense dust cloud, sustaining any injury (excluding eye irritation), fearing being injured/killed, having a relative die in 9/11, having a friend die in 9/11, having a co-worker die in 9/11, being a rescue/recovery worker, losing possessions due to damage, being evacuated from one’s home for at least 48 hours after the attack, and losing a job because of 9/11 [2]. The total number of exposures was then divided into three categories: 0, 1–3, more than 3 exposures.

Mental health conditions included four factors. (1) *Post-9/11 mental health diagnosis* was measured as a dichotomous variable, indicating whether or not participants reported being diagnosed with depression, PTSD, or anxiety disorder other than PTSD by a doctor or other mental health professional after 2001, the year of the attack. If the participant reported any diagnoses having occurred in 2001, we considered the diagnoses to have happened after 9/11 if the respondent did not report any traumatic event other than the WTC attack experienced before 9/11. We used respondents’ wave 2 data to supplement their answers in wave 3 to reduce recall bias. (2) *Number of probable mental disorders*, which indicated symptom severity, was defined as the number of current probable mental disorders in the past 30 days for which respondents met criteria as measured by the PTSD Checklist, Civilian Version (PCL-C, score ≥44) [43], Patient Health Questionnaire Depression Scale (PHQ-8, score ≥10) [44], and Generalized Anxiety Disorder scale (GAD-7, score ≥10) [45]. Within each of the measures, if a certain item was missing, it was coded as “0”; item scores were then summed to see if the respondent met the criteria for the probable disorders based on the cutoff scores indicated above. A variable was then created to indicate the total number of distinct probable disorders present (range 0–3). Individuals who had no response for a certain measure altogether were categorized as not having that disorder. It should be noted that these measures indicated probable PTSD, depression, and generalized anxiety disorder, and were not clinical diagnoses. (3) *Functional impairment* captured the level of difficulty associated with working, taking care of things at home or getting along with others due to PTSD or depression symptoms in the last 30 days from “not difficult at all” to “extremely difficult”. We derived a composite score using the highest level of functional impairment associated with either disorder. Based on the frequency distribution, we dichotomized the variable into “not difficult at all” as not impaired, and “at least somewhat difficult” or higher as impaired. (4) *Poor mental health days* was measured by asking respondents to report the number of days in the past 30 days when their mental health was not good, which was dichotomized as <14 days and ≥14 days. The number of probable mental health diagnoses and the number of poor mental health days were measured within the same time frame and were moderately correlated (Spearman correlation coefficient *r* = 0.46, *p* < 0.0001). We included both in the analyses—the former as an indication of distinct probable disorders and the latter as a more general subjective measure of mental health.

Social support was measured using five items related to perceived social support received in the past 12 months, including how often someone was available to take the respondent to the doctor, have a good time with them, hug them, prepare meals if they are unable to do it themselves, and understand their problems. Each item was rated on a Likert scale from 1 (none of the time) to 5 (all of the time) (recoded as 0–4). Item scores (Cronbach’s alpha = 0.92) were summed to create a total score and divided into quartiles (0–6, 7–11, 12–15, and 16–20). Stress events was defined as experiencing one of the following in the last 12 months: Could not pay for food, housing, or other basic necessities for a period of 3 months or longer; serious family problems involving spouse, child, or parent; took care of a close family member or friend with a serious or life-threatening illness; serious legal problem; and/or lost someone close due to accidental death, murder, or suicide.

Mental health service use was defined as having seen a doctor or health professional or taken any prescription medication for depression, PTSD, anxiety disorder, or nerves, emotions, or other mental health problems or having sought treatment for PTSD symptoms in the past 12 months.

Lacking or disrupted health insurance refers to not having health care insurance at any point in the past 12 months. 

### 2.3. Statistical Analyses

To avoid bias due to exclusion of all missing data and to preserve sample size, we used a “missing” category for all factors with missing values. We examined the distribution of participant characteristics using frequencies and percentages and calculated the prevalence of UMHCN within categories of each factor. We used Chi-square tests to assess bivariate associations between each factor and UMHCN. We also conducted logistic regression analyses, with the outcome of UMHCN, to examine its associations with predictors of interest, adjusting for sets of covariates hierarchically. Mental health condition variables were the main predictors in Model 1, controlling for age, gender, and other sociodemographic as well as WTC exposure variables. Covariates no longer associated with the outcome in the model were removed, with the exception of age and gender, which are important demographic variables in their own right. Model 1 included pre-9/11 mental health diagnosis, number of probable mental disorders, functional impairment due to PTSD or depression, and poor mental health days, as well as the covariates of age, gender, and income. Model 2 included an additional set of predictors related to social support and stressful events. Model 3 extended Model 1 by adding mental health service use and health insurance, since their unique contribution is of interest based on the literature. Model 4 included both social support and stressful events as well as health insurance and service use, controlling for all variables in Model 1. We derived covariate-adjusted odds ratios (aOR) and 95% confidence intervals (CI) from the estimated model parameters to aid interpretation. All analyses were conducted using SAS 9.4 [46].

## 3. Results

Characteristics of the study sample are described in Table 1. The majority of the sample was 25–44 or 45–64 years (79.48%), male (51.54%), had a college degree or higher (57.04%), had a household income of $50,000 or more (55.50%), was employed (64.33%), married or cohabitating (66.60%), and was foreign-born or did not report nativity (70.05%). Close to two-thirds of the participants experienced 1–3 WTC-related exposures (64.29%).

Over 10% had a post-9/11 mental health diagnosis, 22.91% had at least one probable mental disorder, 44.50% had at least some functional impairment due to PTSD or depression symptoms, and 17.70% reported 14 or more poor mental health days in the past 30 days. Over 15% used mental health services in the past 12 months, and approximately 12% had UMHCN in the past 12 months. Among the 281 participants with UMHCN, attitudinal barriers were the most attributed cause for unmet need (68.68%), followed by cost barriers (36.30%) and access barriers (28.47%).

Having UMHCN was significantly associated with several demographic, WTC attack exposure, psychosocial, and clinical factors (Table 1). Higher prevalence of unmet mental health needs was present among those with lower education, lower household income, those who were unemployed, divorced/separated/widowed, and were foreign-born or had unreported nativity. Greater proportion of unmet needs was also noted among those who experienced >3 WTC-related exposures, at least one past year stressful event, and among those with less social support. UMHCN was also more prevalent among those with post-9/11 mental health diagnosis, those with current mental health disorders/impairment, those lacking or having interrupted health insurance, and those who used mental health services in the past 12 months. 

In covariate-adjusted regression analysis, mental health and demographic factors other than gender were associated with greater odds of UMHCN across all four models, although several of these associations were attenuated in models with additional covariates (Table 2). Among the sociodemographic variables, income was significantly associated with UMHCN in all models. Participants younger than 45 years of age had higher odds of unmet needs than those that were older. Adjusting for age, gender, and income, the four variables representing mental health conditions were all significantly associated with UMHCN. Specifically, participants having post-9/11 mental health diagnosis; functional impairment due to PTSD or depression; 14 or more poor mental health days in the past month; and greater number of probable mental disorders had higher odds of unmet mental health needs. Model 2 showed additional effects of stress and social support; the odds of UMHCN were greater among those who experienced at least one stressful event in the past year but lower among those with greater social support. Inclusion of these two factors seemed to slightly attenuate the associations between the outcome and some Model 1 variables, such as income, comorbidity, and functional impairment, but strengthened the effect of age and post-9/11 mental health diagnosis. When controlling for age, gender, income, and mental health conditions in Model 3, the odds of UMHCN greatly increased with the lack of or disrupted health insurance in the past year but were not associated with mental health service use in the past year. The addition of the factors for disrupted health insurance and mental health service use slightly attenuated the associations of all factors in Model 1 with the outcome.

Model 4 included all factors of interest. Greater odds of UMHCN were noted with an increase in number of probable mental disorder (meeting criteria for one vs. no disorder , aOR = 1.59, 95% CI: 1.05–2.39; 2 vs. 0 disorder, aOR = 1.76, 95% CI: 1.08–2.86; 3 vs. 0 disorder, aOR = 2.33, 95% CI: 1.42–3.81), among those with a post-9/11 mental health diagnosis (aOR = 1.61, 95% CI: 1.08–2.38), functional impairment (aOR = 2.36, 95% CI: 1.60–3.48), and having 14 or more poor mental health days (aOR = 1.53, 95% CI: 1.09–2.16). Being aged 25–44 vs. 65 or older (aOR = 1.67, 95% CI: 1.07–2.60) and having a lower vs. higher income (aOR = 1.61, 95% CI: 1.15–2.25) were also associated with greater odds of unmet needs. In Model 4, greater social support was associated with lower odds of unmet need (45–75% reduction), while having experienced stressful events (aOR = 1.32, 95% CI: 0.96–1.81, *p* = 0.08) and lacking or disrupted health insurance in the past year (aOR = 2.37, 95% CI: 1.61–3.48) were both associated with greater odds of UMHCN. Compared to results in Model 2, further controlling for health insurance and service use attenuated the effect of stressful event but had no impact on the association between social support and UMHCN. Similarly, compared to results in Model 3, the addition of social support and stressful events slightly attenuated the effect of health insurance on odds of UMHCN.

## 4. Discussion

A sizeable proportion of Asian Americans had UMHCN 10–11 years after direct exposure to the WTC attack (12%), which is much higher than that reported in other community samples (e.g., 4.5%) [5], indicating greater unmet needs in those affected by this mass trauma more than a decade later. The prevalence of UMHCN is also higher than that reported among the Asian group at wave 2 of the Registry, which was 5–6 years after the disaster (4.4%) [2]. This may be related to delayed manifestation of PTSD symptoms and deferred mental health service use [8]. This high prevalence of UMHCN may also be attributed to dwindling of resources for free services provided by governmental and nongovernmental organizations after the attack (e.g., Project Liberty and American Red Cross 9/11 Fund) [47,48]. This is consistent with our finding that lack or disruption of health insurance had the strongest association with UMHCN among all other factors. 

Among those who reported UMHCN, attitudinal barriers were most common, followed by cost and access barriers. This is in line with other studies that have reported attitudinal barriers (e.g., acceptability) as more likely causes of unmet needs compared to other tangible barriers, such as accessibility and availability of mental health services [5]. Within those who attributed their unmet need to attitudinal barriers, “prefer to manage myself” was the most endorsed item (62.1%, not shown in tables), which may indicate the desire to keep the issue private and potentially related to being “afraid to ask for help/of what others would think” (25.4%). Both may be manifestations of fear of stigma. Other common reasons like “didn’t get around to it/didn’t bother” (37.8%), and “don’t think anything could help” (34.2%) may reflect the tendency to undervalue treatment due to the lack of knowledge about mental illness and its treatment. These figures are disconcerting since these Asian Americans had been directly exposed to the WTC attack and should have heard of its mental health impact from the media. It is possible that they no longer relate their mental health need to the disaster a decade later. 

In terms of factors associated with UMHCN, it is noteworthy that those who lacked or had disrupted insurance in the past year had more than twice the odds of having unmet needs than those who did not. However, this was consistent with greater odds of UMHCN for individuals from household income of <$50,000 than those of higher income, and the significant association persisted throughout the hierarchical regression models. Close to 40% of Asian participants reported household income below $50,000, and 15% were unemployed; thus, they may not be able to afford or willing to pay for services to meet their mental health needs. This is consistent with cost being the second most common reason for UMHCN after attitudinal barriers. Asians at the prime years of their lives (25–44) had significantly higher odds of having UMHCN, which may reflect the challenges of financial and family-related responsibilities [49,50]. 

The association between mental health conditions and UMHCN noted in this study is expected, since less favorable mental health conditions would increase mental health care needs and thus lead to a higher chance that the need is not being met. The finding is consistent with reports from previous studies based on the Registry across race [27] and other data [7]. These mental health factors were all significant in their association with the outcome throughout the four models. It is of interest to note that among the mental health factors, functional impairment from PTSD or depression had the strongest association with UMHCN, with even higher odds compared to having comorbidity of three probable mental disorders, which indicated severe symptomatology. This may reflect Asians’ tendency to dismiss their mental health need unless it interferes with their daily functioning [14]. Further, those with a low income and those who were unemployed may also be less able to compensate for their inability to perform expected roles at work and at home to sustain themselves and their families, which may cause them to recognize their UMHCN. The association between post-9/11 mental health diagnosis and UMHCN suggests that Asian Americans who had previous contact with mental health professionals may have been more aware of their mental health needs and when those needs were not met, whether or not they continued treatment. 

Asian Americans with stressful life events in the past year, including financial and health challenges to themselves or families, could tax their coping resources and thus increase mental health issues and thereby heighten the awareness of their UMHCN. However, social support may have alleviated unmet needs by reducing access barriers to mental health care by providing practical assistance [5] such as child care or meal preparation when individuals sought mental health care, or accompanying them to see the doctor. At the same time, the relatively higher mental health stigma noted among Asian Americans in general [51] may have attenuated the positive impact of social support on UMHCN in this study. The fact that attitudinal barriers were the most common reasons for UMHCN among Asians may also indicate fear of stigma within the social network should they seek mental health services. Thus, exploration of the perception of stigma within the community or within social networks and its relation to unmet needs is an important area of future inquiry in disaster-affected Asian communities.

The lack of association between previous mental health service use and UMHCN found in this study contrasts with findings from most studies in which previous consumers were more likely to indicate unmet needs [5,27], potentially due to dissatisfaction with the quality of treatment, recognition of potential help that they cannot access, or limitations of services in providing desired help after being exposed to the formal mental health care system [5]. This nonsignificant finding was unexpected also because it contradicts with our expectation that there would be higher dissatisfaction with mental health treatment among Asians due to poorer quality of service [4] and lower therapeutic alliance due to cultural and linguistic barriers [34,35]. It is possible that the smaller proportion of Asian Americans having received mental health services (e.g., compared to the non-Hispanic white group, 15.7% vs. 26.6%; the authors, in press) limited statistical power. However, the tendency of Asian Americans to terminate treatment prematurely [13] suggests that they may be less exposed to the mental health care system, hence less likely to recognize the range of available treatment that is inaccessible to them, and therefore less likely to indicate UMHCN [5]. Moreover, lower mental health literacy could also prevent them from recognizing an unmet need [25], also called “unperceived unmet need” [6]. Furthermore, their mental health need might have been satisfied through effective mental health services such as those provided by the WTC Health Program [52], or the use of alternative healing methods such as traditional Chinese medicine [53] or other folk healing within their unique communities [54].

### 4.1. Study Limitations and Strengths 

The findings of this study should be considered in light of the study’s limitations. The study was unable to capture ethnic diversity among Asian Americans in the sample. While there are similarities among Asian ethnic groups, it is possible that the various factors we examined may be associated with UMHCN differentially among the Asian subgroups [55,56,57]. Further, since UMHCN is captured by self-report without any clinical record for confirmation, which could indicate individuals’ mental health care needs and premature termination; when individuals are unaware of their mental health need, this could cause underestimation of its prevalence. This is particularly an issue for many Asian Americans who have limited mental health knowledge and the tendency to ignore their psychological distress [13,14]. The prevalence of unmet needs may also be underestimated due to social desirability bias, in which participants avoided reporting mental health care needs to avoid stigma [25]. The lack of information on the duration of previous treatment, number of visits to mental health service, type of providers seen, or satisfaction with services also limited our ability to assess the impact of previous mental health service use on UMHCN. Due to sample size limitations, we were also unable to assess specific factors associated with UMHCN due to attitudinal, cost, and access barriers separately. Further inquiry in this regard could better inform strategies to ameliorate the effects of these specific barriers. Finally, since the investigation is based mainly on cross-sectional data from wave 3, causal relationships could not be established. 

Despite these limitations, this study is the single largest investigation of UMHCN among Asian Americans who were directly exposed to the WTC attack, and the first large-scale study examining Asians’ subjective perception of their UMHCN after a massive trauma. The direct assessment of individuals’ subjective perception of unmet needs has the advantage of capturing a potential service gap without assuming that a mental health service need is satisfied with service use alone. The comprehensive recruitment and follow-up efforts of the Registry through the use of Asian-language interviewers and outreach efforts [42] enabled the inclusion of individuals who were not proficient in English, which constituted a large proportion of the Asian American population (75%) [17]. The longitudinal design of the study allows for examination of longer term unmet needs, which is critical in light of the often delayed manifestation of PTSD symptoms and service use [8]. 

### 4.2. Study Implications

Our finding suggests that a substantial proportion of Asian Americans have UMHCN a decade after the WTC attack, which was even greater than that noted 5 years prior in another study [2]. This may indicate that continued outreach efforts to this community are necessary to provide treatment options. During the years after the disaster, with heightened awareness of its impact through media coverage and outreach efforts from government and nongovernmental organizations, mental health services were made more readily available. However, with time, the number of organizations providing free services and public awareness of some continued programs declined, potentially increasing practical barriers such as accessibility and cost. Thus, programs such as the Registry’s Treatment Referral Program which reaches out to enrollees who lived or worked near the disaster area and reported a physical and/or mental health symptom should be continued long term [58]. Since those with worse current mental health conditions and previous diagnoses have higher odds of UMHCN, they should also be targeted for interventions that bring them to treatment and to improve treatment adherence. To ensure that culturally competent mental health service providers are available is also important [21] to increase continuation and satisfaction with services in fully meeting their needs. 

Since the largest proportion of UMHCN among Asian Americans exposed to the WTC attack was attributed to attitude about mental illness and its treatment, a “subjective chosen unmet need” [6], public mental health education may well be key to address unmet needs with this population. Increasing knowledge about the cause, manifestation, and course of mental illness could help to reduce stigma, increase awareness of individuals’ mental health needs, and promote appropriate expectations of the treatment process so as to encourage service use and reduce dropout. These public education efforts should be tailored to the languages, needs, and cultures of the various minority communities [21]. The importance of post-disaster large-scale public mental health education and outreach efforts is also relevant for future disasters. 

Another important implication of this study’s findings is the strong role of health insurance, income, and employment in unmet needs. This suggests that free mental health care related to the WTC attack should be continued. The WTC Health Program under the WTCHR has provided health and mental health services to individuals directly exposed to the disaster, and outreach efforts to affected individuals through the Treatment Referral Program continues to date. Enrollees who lived or worked in the 9/11 disaster area and reported a physical and/or mental health symptom were encouraged to seek care from 9/11-specialty providers [52]. This program should be continued given the considerable UMHCN noted in this study among the Asians, who may not be able to afford needed services. Although the James Zadroga 9/11 Health and Compensation Act enacted in 2010 was extended through 2090 [10], the recent report on the September 11th Victim Compensation Fund released in February 2019 is concerning, which revealed funding insufficiency and needs for reduced awards due to greater claim than anticipated [59]. Congress would need to consider providing more resources for free services through adequate compensation of treatment expenses. Given the long-term nature of mental illnesses, especially PTSD, this is an important consideration.

With the persistent and striking underutilization of mental health services among Asian Americans, the value of the subjective account of UMHCN to better understand the actual demand for mental health services and the perceived adequacy and quality of treatment [7] becomes more important, and research along this line is necessary. In future studies, ethnicity within the Asian American population should be disaggregated to gain a more nuanced understanding of unique unmet needs and their specific barriers so that improvement in outreach efforts and mental health treatment could be made to ameliorate the disparities in mental health services [6]. Further, for those who have used mental health services before and still find their needs unmet, we need to understand in greater depth the “dosage” of services received, whether or not it was premature termination, and reasons for their dissatisfaction. More detailed exploration of the stigma Asians ascribe to mental health problems and treatment should be explored so that relevant community mental health education can more effectively address any misunderstanding.

## 5. Conclusions

The prevalence of UMHCN among Asian Americans directly exposed to the WTC attack remained sizeable a decade after the attack, and the cause of unmet need was attributed most frequently to attitudinal barriers, followed by cost and access barriers. Thus, continued outreach efforts to Asian communities through public mental health education to increase knowledge about mental illness and the value of and options for mental health treatment, and to destigmatize the illness and treatment, is of paramount importance. Free and convenient access to mental health services to eligible individuals through adequate allocation of public funding is crucial to ensure that issues of cost and access will also not prevent individuals from addressing their mental health needs. Further exploration of subjective perception of UMHCN to provide information on the actual demand for mental health services and ways to improve treatment among Asian Americans is also warranted.

## Figures and Tables

**Table 1 ijerph-16-01302-t001:** UMHCN of Asian American Participants and their Characteristics (*N* = 2344).

Factors		N	%	UMHCN N	UMHCN %	*p*-Value
UMHCN (DV)	Yes	281	11.99			
Reasons for UMHCN by type	Attitudinal	193	68.68			
	Cost	102	36.30			
	Access	80	28.47			
**Demographics**						
Age (years)	18–24	78	3.33	8	10.26	0.17
	25–44	736	31.40	98	13.32	
	45–64	1127	48.08	119	10.56	
	65+	403	17.19	56	13.90	
Gender	Male	1208	51.54	140	11.59	0.442
	Female	1076	45.90	136	12.64	
	Missing	60	2.56	5	8.33	
Education	<College graduate	902	38.48	133	14.75	0.001
	≥College graduate	1337	57.04	136	10.17	
	Missing	105	4.48	12	11.43	
Income	<$50,000	915	39.04	171	18.69	<0.001
	≥$50,000	1301	55.50	95	7.30	
	Missing	128	5.46	15	11.72	
Employment status	Employed	1508	64.33	149	9.88	<0.001
	Unemployed	815	34.77	121	14.85	
	Missing	21	0.90	11	52.38	
Marital status	Married/cohabiting	1561	66.60	159	10.19	<0.001
	Divorced/separated/widowed	324	13.82	54	16.67	
	Never married	439	18.73	63	14.35	
	Missing	20	0.85	5	25.00	
Nativity	US born	702	29.95	77	10.97	0.002
	Foreign-born/unreported	1642	70.05	204	12.42	
**WTC Attack Exposure**						
WTC Exposure	0	452	19.28	40	8.85	<0.001
	1–3	1507	64.29	167	11.08	
	≥4	385	16.42	74	19.22	
**Mental Health Condition**						
Post-9/11 mental health diagnosis	No	2103	89.72	212	10.08	<0.001
	Yes	241	10.28	69	28.60	
No. of probable mental disorders (PCL, PHQ-8, GAD-7)	0	1807	77.09	129	7.14	<0.001
	1	243	10.37	49	20.16	
	2	129	5.50	39	30.23	
	3	145	6.19	60	41.38	
	Missing	20	0.85	4	20.00	
Functional impairment due to PTSD/depression	Not impaired	1008	43.00	45	4.46	<0.001
	Some impairment	1043	44.50	222	21.28	
	Missing	293	12.50	14	4.78	
Poor mental health days	<14	1874	79.95	155	8.27	<0.001
	≥14	415	17.70	117	28.19	
	Missing	55	2.35	9	16.36	
**Social Support and Stress**						
Social support (Cronbach coefficient alpha=0.92)	0-6	504	21.50	121	24.01	<0.001
	7-11	481	20.52	71	14.76	
	12-15	527	22.48	39	7.40	
	16-20	742	31.66	29	3.90	
	Missing	90	3.84	21	23.33	
Stressful events in the last 12 months	No	1793	76.49	163	9.09	<0.001
	Yes	551	23.51	118	21.42	
**Mental Health Service Use and Health Care Resources**						
No health insurance at any point within 12 months	No	2031	86.65	189	9.31	<0.001
	Yes	227	9.68	67	29.52	
	Missing	86	3.67	25	29.07	
Service use in past 12 months	No	1562	66.64	158	10.12	<0.001
	Yes	369	15.74	99	26.83	
	Missing	413	17.62	24	5.81	

UMHCN = unmet mental health care need, DV = dependent variable, WTC = World Trade Center, PCL = PTSD checklist, PHQ-8 = Patient Health Questionnaire, GAD-7 = generalized anxiety disorder, PTSD = posttraumatic stress disorder. All *p*-values were from Chi-square tests using data with missing category excluded.

**Table 2 ijerph-16-01302-t002:** Covariate-adjusted odds ratio (aOR) and 95% confidence interval (CI) derived from logistic models for UMHCN (*N* = 2344).

Factors		Model 1	Model 2	Model 3	Model 4
		aOR (CI)	aOR (CI)	aOR (CI)	aOR (CI)
Demographics					
Age (years)	18–24 25–44 45–64 65+	2.08 (0.63–6.90) 1.70 (1.12–2.57) 0.90 (0.62–1.32) 1	2.95 (0.85–10.23) 1.91 (1.25–2.93) 0.88 (0.59–1.29) 1	1.75 (0.52–5.91) 1.41 (0.92–2.18) 0.79 (0.53–1.17) 1	2.47 (0.70–8.68) 1.67 (1.07–2.60) 0.78 (0.52–1.17) 1
Income	≥$50,000 <$50,000	1 2.61 (1.92–3.54)	1 2.03 (1.48–2.79)	1 2.03 (1.47–2.80)	1 1.61 (1.15–2.25)
Mental Health Conditions					
Post-9/11 mental health diagnosis	No Yes	1 1.73 (1.21–2.49)	1 2.49 (1.84–1.27)	1 1.52 (1.04–2.23)	1 1.61 (1.08–2.38)
No. of probable mental disorders (PCL, PHQ-8, GAD-7)	0 1 2 3	1 1.79 (1.21–2.66) 2.18 (1.36–3.48) 2.88 (1.81–4.59)	1 1.68 (1.12–2.51) 1.85 (1.14–2.98) 2.46 (1.52–3.98)	1 1.70 (1.14–2.55) 2.08 (1.29–3.35) 2.71 (1.68–4.38)	1 1.59 (1.05–2.39) 1.76 (1.08–2.86) 2.33 (1.42–3.81)
Functional impairment due to PTSD or depression	Not impaired Some impairment	1 3.04 (2.09–4.43)	1 2.40 (1.64–3.53)	1 2.94 (2.01–4.30)	1 2.36 (1.60–3.48)
Poor mental health days	<14 ≥14	1 1.68 (1.21–2.34)	1 1.63 (1.16–2.27)	1 1.57 (1.12–2.20)	1 1.53 (1.09–2.16)
Social Support and Stress					
Social support	0–6 7–11 12–15 16–20		1 0.58 (0.41–0.83) 0.38 (0.25–0.57) 0.25 (0.16–0.40)		1 0.55 (0.38–0.79) 0.37 (0.24–0.57) 0.25 (0.16–0.41)
Stressful events in the last 12 months	No Yes		1 1.44 (1.06–1.96)		1 1.32 (0.96–1.81)
Mental Health Service Use and Health Care Resources					
No health insurance at any point within 12 months	No Yes			1 2.47 (1.69–3.61)	1 2.37 (1.61–3.48)
Service use in past 12 months	No Yes			1 1.26 (0.89–1.77)	1 1.31 (0.92–1.86)

PCL = PTSD Checklist, PHQ = Patient Health Questionnaire, GAD = Generalized Anxiety Disorder, PTSD = posttraumatic stress disorder. Each model includes variables for which there are estimates provided in that column.

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
