# Peer review of "Unmet Mental Health Care Needs among Asian Americans 10–11 Years After Exposure to the World Trade Center Attack"

_ijerph, 2019, doi:10.3390/ijerph16071302_

Round 1
Reviewer 1 Report
This paper identifies and tests various predictors of unmet mental health care needs among Asians 10-11 years following exposure to the WTC attacks. This research question is important because this population may face additional barriers to receiving mental health care, identifying such barriers is an important contribution to the public health post-disaster literature. It examines a unique study population derived from a valuable database on individuals exposed to the WTC attacks on 9/11. The paper is generally well written, with opportunities for improved clarity in the text. I have a few concerns, described below.
Major concerns
Methods:
• In the creation of the predictors, a discussion of how missing scores or categories were included is missing. Using the PCL, PHQ, and GAD surveys, participants could have had missing surveys completely or missing answers in the generation of the scores. How were these handled in the creation of the survey scores and in the 0-3 summary score?
• Similar to the above issue, there could be considerable overlap in the mental health condition predictor and the number of probable mental disorders. A deeper explanation of how the PCL, PHQ, and GAD represent symptomology and not necessarily a clinical diagnosis in addition to the explicit time frame these two predictors represent would help the reader to understand. Authors may also consider examining how correlated these two predictors are.
• Authors included a missing category for nearly all of their predictors in the models yet did not discuss the rationale behind this (presumably to preserve the already small sample size) or a discussion of any of the observed results for the missing categories. For example, under the number of probable mental disorders section in Table 2, the missing category has some very high ORs. What does this mean and who are these individuals. Authors should consider removing this missing category or provide explicit rationale for the inclusion and a discussion of who these individuals are in the discussion section.
• The text regarding the model building process in the statistical analysis section could be further clarified. It’s unclear to the reader at the end of the paragraph exactly which variables are included in which models, and the tables don’t include this information either. For example, “control variables no longer associated with the outcome in the model were removed” is not clearly defined so the reader does not know what the final variables in each model were. The use of the words “control variable”, “predictors”, “variables”, “and covariates” could be more consistent to further aid the reader.
Discussion:
• P. 12, Paragraph 2 and P. 13 Lines 414-421: There is no discussion of the care provided by the WTC Health Program. Mental health care needs may have been satisfied by the care provided via this (or similar) programs, a possibility that should be mentioned as to not discredit or devalue the Health Program as a resource that has been available to this population. This program, which includes a wide variety of mental health disorders as qualified conditions, may be more important to meeting UMHCN than programs like the Victim’s Compensation Fund, as such, it should be included in the discussion.
Overall:
• There are several references to the general Asian population being from a lower socioeconomic status. Many of the sentences are supported by very old references or references which refer to a specific ethnic category. And generally, the demographic characteristics of this sample don’t support the statements that this entire group is of “low socioeconomic status”: 57% have a college education or more, 66% are married, and 55% have incomes above $50,000 (50,000 is still more than the median US wage). Would suggest softening the language around SES as being the primary driver behind these associations.
• P. 2, line 73 “…but have less resources to sustain themselves and their families when they cannot perform their expected roles, especially work which is highly valued among Asians”
• P.2, line 81 “strong impact on Asian Americans given their relatively lower socioeconomic status”
• P. 11, line 328 “Further, their low socioeconomic status also poses more limitations to compensate for their inability to perform expect roles at work and at home to sustain themselves and their families, which causes them to recognize their UMHCN”
Minor concerns
Introduction
• P.2, Line 69: Unclear what “without formal evaluation” is referring to
• P.2, Line 87-88: Unclear what “result of malingering bad thoughts and personality weakness” means, the references supporting this sentence are also quite old
• P. 3, Line 122-125: Unclear what “raising the awareness of their subjective UMHCN for Asians” means
Methods:
• P. 3, line 130-131: Please confirm this is the correct funding source for the WTCHR
• P. 3, Line 139: Clarify at what wave or time point age 18 was defined (time of 9/11, Wave 1, etc)
• P. 4 UMHCN definition: Please clarify if the attitudinal, cost, and access barriers were mutually exclusive categories or if you could check all that apply and how this was handled in the creation of these groups
• P. 4, Please further clarify how functional impairment was operationalized in the models. It’s unclear how the composite score was created and what it means by “grouped them based on presence of any difficulty”
• P. 4, Line 154: Please clarify which wave marital status and household income were defined from, and perhaps add the categories for all of these covariates
Results:
• The results section was extensive and very repetitive of the tables. For example, lines 233-247 highlight every variable included in table 1 with the proportions and p-values. A summary or highlight of important covariates/associations would be preferred over repeating all the information contained in the tables.
Discussion
• P. 12, Lines 339: The wording of this sentence needs to be clarified as “higher mental health stigma noted in this population” is not referring to the current sample but to a finding in an old paper.
• P. 12, Lines 340-343: This sentence is confusing and seems to state the opposite of what the beginning of the paragraph stated. Please clarify what authors mean by exploration of the “negative impact of social support on unmet need” is.
Tables:
• Should be stand alone with spelled out abbreviations, footnotes including which covariates were included in each models, what bold signifies
• Table 1: Presenting the results by N % and then % (N) is very confusing to the reader, would suggest consistently reporting N % in the tables.
• Table 1: Table formatting came through unclear, but it seems the income referent level is opposite to what is referred to in the text? Table shows <$50,000 is referent; line 311-313 states “greater odds of UMHCN for individuals from household income of <$50,000 than those of higher income…” which is opposite of the table findings which show those with higher incomes have greater odds of UMHCN compared to lower incomes.
Author Response
Response to Reviewer 1:
Our team very much appreciates the thoughtful and detailed feedback from the reviewer and addressed and accepted all of the suggestions. Here are our responses below, and all the changes were marked in the manuscript using track changes:
Major Concerns:
Methods:
1. In the creation of the predictors, a discussion of how missing scores or categories were included is missing. Using the PCL, PHQ, and GAD surveys, participants could have had missing surveys completely or missing answers in the generation of the scores. How were these handled in the creation of the survey scores and in the 0-3 summary score?
Response: Discussion of how missing scores/categories were handled in creating survey scores and the 0-3 summary variable is now included under measures of mental health conditions on p.2 para.3. The PCL, PHQ-8 and GAD-7 represent symptomatology of the disorders but not clinical diagnoses was also indicated.
2. Similar to the above issue, there could be considerable overlap in the mental health condition predictor and the number of probable mental disorders. A deeper explanation of how the PCL, PHQ, and GAD represent symptomology and not necessarily a clinical diagnosis in addition to the explicit time frame these two predictors represent would help the reader to understand. Authors may also consider examining how correlated these two predictors are.
Response: Regarding the correlation between mental health conditions, a Spearman correlation was calculated for the association between the number of probable mental health diagnoses and the number of poor mental health days and reported on p.5 para.1, resulting in a moderate correlation between the two measures (r=0.46).
3. The text regarding the model building process in the statistical analysis section could be further clarified. It’s unclear to the reader at the end of the paragraph exactly which variables are included in which models, and the tables don’t include this information either. For example, “control variables no longer associated with the outcome in the model were removed” is not clearly defined so the reader does not know what the final variables in each model were. The use of the words “control variable”, “predictors”, “variables”, “and covariates” could be more consistent to further aid the reader.
Response: The rationale for including a missing category for all predictors is explained on p.5 para.5 under statistical analyses, which was to avoid possible bias due to the exclusion of all missing data and to preserve the already small sample size. Frequencies and percentages for missing categories are provided in Table 1. Table 2 presented OR comparing odds of UMHCN between reference and other categories for each factor adjusting for other factors in the same model. As suggested, we removed the missing category from Table 2.
4. The text regarding the model building process in the statistical analysis section could be further clarified. It’s unclear to the reader at the end of the paragraph exactly which variables are included in which models, and the tables don’t include this information either. For example, “control variables no longer associated with the outcome in the model were removed” is not clearly defined so the reader does not know what the final variables in each model were. The use of the words “control variable”, “predictors”, “variables”, “and covariates” could be more consistent to further aid the reader.
Response: In the statistical analyses section on p.5, para. 4, we also added a description of the variables retained in Model 1. We kept the term “predictor” in the discussion as it referred to the variable of interest but we changed the term “control variable” to “covariates” for consistency, as suggested.
Discussion:
5. P. 12, Paragraph 2 and P. 13 Lines 414-421: There is no discussion of the care provided by the WTC Health Program. Mental health care needs may have been satisfied by the care provided via this (or similar) programs, a possibility that should be mentioned as to not discredit or devalue the Health Program as a resource that has been available to this population. This program, which includes a wide variety of mental health disorders as qualified conditions, may be more important to meeting UMHCN than programs like the Victim’s Compensation Fund, as such, it should be included in the discussion.
Response: The value of programs like the WTC Health Program to satisfy this population’s mental health care needs is mentioned on p.11 para.2, and the importance of its continuation is discussed on p.12 para.5.
Overall:
6. There are several references to the general Asian population being from a lower socioeconomic status. Many of the sentences are supported by very old references or references which refer to a specific ethnic category. And generally, the demographic characteristics of this sample don’t support the statements that this entire group is of “low socioeconomic status”: 57% have a college education or more, 66% are married, and 55% have incomes above $50,000 (50,000 is still more than the median US wage). Would suggest softening the language around SES as being the primary driver behind these associations.
• P. 2, line 73 “…but have less resources to sustain themselves and their families when they cannot perform their expected roles, especially work which is highly valued among Asians”
• P.2, line 81 “strong impact on Asian Americans given their relatively lower socioeconomic status”
• P. 11, line 328 “Further, their low socioeconomic status also poses more limitations to compensate for their inability to perform expect roles at work and at home to sustain themselves and their families, which causes them to recognize their UMHCN.”
Response: We agree that we should be cautious not to overstate the low socioeconomic status (SES) of the Asian group in this study, and we revised the language around SES accordingly, e.g. on p.2 para.2, and p.10 para.4.
Minor Concerns:
All the minor concerns were address: some points were clarified, required information was given, and some references were updated, where indicated, by track changes. In the results section, findings were more succinctly summarized as suggested. Footnotes in the table were also added, as suggested.
We hope that the reviewer finds the responses satisfactory. Thank you very much for your attention and assistance.

Reviewer 2 Report
Line 50 - I suggest the sentences ‘This study..’ be inserted and paraphrased at the end of Introduction section
Line 110 - there’s typo ‘unrealistic such a quick fix..’
Line 129 - How many study participants is being considered for this study? Please explicitly mention the number
Line 173 - PHQ-8 typo
Line 318 - perhaps this sentence would be better with relevant citation(s)
Line 424 - this sentence can be integrated with the previous sentence
Author Response
Response to Reviewer 2
Our team very much appreciates the reviewer’s careful editorial suggestions and made changes accordingly with track changes in the manuscript.
1. The clear restatement of the study purpose was added to the end of the introduction section on p.3 para.3.
2. The typo on “…unrealistic such as a quick fix…” was corrected.
3. The number of participants was more clearly explained and given on p.3 last paragraph.
4. PHQ8 was changed to PHQ-8, and GAD7 to GAD-7.
5. Pertinent reference was given related to “Asians at the prime years of their lives (25-44) had significantly higher odds of having UMHCN, which may reflect the challenges of financial and family-related responsibilities.” on p.10 para.3 at the end of the paragraph.
6. The suggested integration of the two sentences on p.2 last paragraph was made: “With the persistent and striking underutilization of mental health services among Asian Americans, the value of the subjective account of UMHCN to better understand the actual demand for mental health services and the perceived adequacy and quality of treatment [7] becomes more important, and research along this line is necessary”
We hope that the reviewer finds the responses satisfactory. Thank you very much for your attention and assistance.
